# Exploring the Information Sources Consulted by Doctors at the Point of Care in Four Selected South African Referral Hospitals

**DOI:** 10.3390/healthcare12010008

**Published:** 2023-12-19

**Authors:** Nombulelo Chitha, Nkanyiso Ntsele, Sikhumbuzo A. Mabunda, Itumeleng Funani, Buyiswa Swartbooi, Onke Mnyaka, Jahman Thabede, Ruth Tshabalala, Guillermo Alfredo Pulido-Estrada, Sibusiso Nomatshila, Wezile Chitha

**Affiliations:** 1Health Systems Enablement and Innovation, University of the Witwatersrand, Johannesburg 2050, South Africabswaartbooi@witshealth.co.za (B.S.); omnyaka@hsei.co.za (O.M.); jthabede@hsei.co.za (J.T.);; 2Department of Public Health, Walter Sisulu University, Mthatha 5117, South Africa; 3School of Population Health, University of New South Wales, Sydney 2052, Australia

**Keywords:** information sources, doctors, point of care, teaching hospitals

## Abstract

Background: To provide an understanding of the clinical information sources consulted by teaching or referral hospital-based doctors in four South African provinces. Methods: A quantitative cross-sectional survey design was used. To identify provinces, hospitals, and participants, simple random sampling was adopted. This study targeted a total of 276 doctors from all the four hospitals working across different departments within the hospitals. This study was conducted in four selected South African public referral/teaching hospitals in four different provinces, namely Nelson Mandela Academic Hospital in the Eastern Cape province; Witbank Hospital in Mpumalanga province; Robert Mangaliso Sobukwe Hospital in Northern Cape province and lastly, Pietersburg Hospital in Limpopo province. Results: Overall, 221 doctors were surveyed. Doctors relied more on colleagues as formal and informal sources of information. They seldomly relied on newspapers, reference, and library books, or used hospital computers to access the internet. They seldomly attended training workshops organised by the district or provincial office. Protocols and clinical guidelines which are kept in the hospitals and easily accessible were often (27.9%) or always (51.1%) used. Conclusions: Teaching hospitals need to strengthen information resources to ensure that even when colleagues are used as an information source, they are an accessible means to validate the correctness of the information provided.

## 1. Introduction

In healthcare, doctors are responsible for using the knowledge accumulated over time to meet patients’ needs [1]. While seeing patients, doctors keep most of the information they need at the back of their heads [2]. However, some doctors may have outdated information in memory, because new information may not have been acquired, which may lead to medical errors or sometimes resulting in limited capability of handling patients with rare conditions [1,3,4]. In addition to that, it becomes difficult for doctors to practice quality medicine unless they constantly update their knowledge and search for information to help them treat patients [4]. Uninformed or delayed decisions might be made if their information needs remain unmet [4]. Medical errors and patient outcomes can be significantly improved by using appropriate information sources, and it is evident that medical practice is impacted by this in a significant way [5].

It is essential that health professionals have open access to understandable, reliable, and useful clinical information sources [6]. Almost every health worker in middle- to high-income countries now relies on Information and Communication Technology (ICT) to carry out their work [7]. Due to the use of advanced measurement technologies, there is a greater amount of clinical information, and by using the internet available around the globe, information can be accessed at any place and time of the day, provided there is an internet coverage and a working device [5].

Online information and knowledge of the internet are becoming progressively popular among health professionals as the means of obtaining and sharing information [4,5,8]. In addition to facilitating the sharing of health information, the internet allows health professionals to monitor and track diseases, it has also made collaboration, communication, and interaction between doctors easier and possible [4,9]. Even so, some doctors still face several barriers in accessing information to be used for decision making at the point of care [10]. For example, lack of ICT skills, high subscription fees to journals, lack of confidence, inadequate resources, and lacking time to learn how to make use of ICTs [5,10]. In cases where physicians do not have sufficient time and search skills, they can gain valuable skills from librarians in the retrieval, management, and evaluation of information sources which influence point of care decisions and practice [4,11,12]. In addition to providing access to information in underserved areas, they can support the efforts of health professionals to become lifelong learners [13]. The libraries of most universities and hospitals in China, Europe and United States provide clinicians with access to current medical literature, but, unfortunately, many clinicians especially in low and middle-income countries (LMICs) do not have access to up-to-date medical literature through their hospital libraries or universities, even though they can benefit when the sources are openly accessible [14].

In spite of the barriers, there are various sources of information that are consulted by doctors to satisfy their information needs [1,15,16]. Information sources that are mostly used by doctors include: evidence-based medicine, libraries, electronic databases (e.g., MEDLINE, Cochrane library and Google Scholar) textbooks, systematic journal reviews, internet and drug reference books. Other information sources accessed include continued medical education, and human sources such as physicians and colleagues, as a result of their experience and knowledge (perceived or real) [1,4,17].

The utilisation of colleagues such as pharmacists is also found to be a useful source in ensuring that medication prescriptions are correct [16,18]. A Nigerian study by Oshikoya et al. [19] found that many LMICs normally have one Pharmaceutical Sales Representative (PSR) for every five doctors, making them the main primary source of information on medicines [19]. As such the attitude of medical students towards prescribing have been positively affected by training them to interact appropriately with PSRs [19]. In addition, since evidence-based medicine increasingly became popular, journals are also receiving more attention due to their ability to provide doctors with knowledge that can be applied directly in the care of their patients for positive outcomes [4]. Doctors have been reported to find printed materials, such as textbooks, to be unreliable because information in these sources might be outdated since medical practices should follow the most current guidelines towards patient management [4].

Since the late 1990s there has been a lot of reliance on the internet sources among health sciences students and is projected to be a reliable source of information for patient care in future [20,21,22,23]. Whereas health professionals mostly relied on textbooks and colleagues before the year 2000 [21], most doctors and pharmacists have now been found to update their patient care knowledge via the internet including the use of search engines to find drug information [22,23]. It has been found that doctors often interact with colleagues who serve as an informal and formal primary source of information in the hospital setting [16]. In addition, clinical decisions are also perceived to be influenced by them [6,16].

Many African studies have explored other aspects of information seeking, yet there’s a lack of theoretical knowledge on the clinical information sources that South African doctors consult, and the problems they could experience while trying to access the information sources. However, with the advent of information technology, current clinical information sources can now be easily accessed by clinicians in most LMICs [6]. Yet, some health information sources remain underutilised despite being readily available on the internet [24]. This may be due to the difficulties encountered while accessing information sources especially for doctors in South African teaching hospitals. Therefore, a deeper understanding of how South African referral hospitals’ evidence-based practice is influenced by clinical information sources was considered essential for this study to contribute new findings to the body of information seeking behaviour in South Africa. This study therefore sought to determine the clinical information sources consulted by referral hospital-based doctors, for making decisions at the point of patient care in four South African provinces. This information will be shared with policymakers to enhance the strengthening of health systems and teaching hospitals in South Africa.

## 2. Materials and Methods

### 2.1. Study Design

This study used a cross-sectional survey design using a quantitative methodology. Hospitals and provinces were determined by simple random selection. A validated questionnaire will be given to doctors who were chosen through stratified random selection. A standardised, self-administered questionnaire will be used in the investigation. Robert Mangaliso Sobukwe Hospital (Northern Cape province), Pietersburg Hospital (Limpopo province), Witbank Hospital (Mpumalanga province), and Nelson Mandela Academic Hospital (Eastern Cape province) were the sites of choice. The methods used in this study have been described by Chitha [25], and briefly, a quantitative, descriptive cross-sectional study design was used to answer the objectives.

### 2.2. Participants and Setting

Private and public health systems coexist side by side in South Africa. The great majority of people are served by the public health sector. This study focused on Tertiary and Central hospitals. The National Department of Health, province health departments, and local health departments each have different levels of authority and responsibility for providing services. Over 200 private hospitals and over 400 governmental hospitals are available. The largest regional hospitals are directly managed by the provincial health departments. District management oversees primary care facilities and District hospitals. The 10 Central hospitals are directly managed by the Department of Health at the national (federal) level. Among the nine provinces, the Eastern Cape has 91 public hospitals, of which one is a Central hospital and three are Tertiary hospitals; Limpopo has 42 public hospitals of which one is a Tertiary hospital; Mpumalanga has 33 public hospitals of which two are Tertiary hospitals; and the Northern Cape has 16 public hospitals of which one is a Tertiary hospital. Given these, one Tertiary hospital was purposively chosen from each of the three provinces, with the only Central hospital selected in the Eastern Cape province.

Due to resource limitations, four South African rural provinces (Limpopo (LP), Eastern Cape (EC), Northern Cape (NC) and Mpumalanga (MP) were randomly selected for this study. A referral hospital was purposely selected in each of the four provinces, Nelson Mandela Academic Hospital (NMAH) in the EC; Witbank Hospital (WH) in MP; Robert Mangaliso Sobukwe Hospital (RMSH) in NC and lastly; Pietersburg Hospital (PH) in LP. Stratified random sampling of doctors was carried out between the 14 March and 15 September 2022. Study participants were doctors from all the hospitals’ departments. The sample size has been previously discussed by Chitha [25] in the literature, a brief review of those discussions is provided here. Considering the number of doctors in each hospital, sample sizes were proportionally weighted to reach the target of 276 doctors.

### 2.3. Measurements

The questionnaire (Appendix A) was adapted from a validated instrument that was previously used to ascertain the information behaviour of nurses and doctors of one rural South African district [25]. A validated and standardised, structured self-administered questionnaire was used, and asked questions on demographic characteristics, responsibilities, preferred information sources and reasons for choosing an information source. The Cronbach’s alpha had good internal consistency (0.865) for the 25 items assessed on the three and five item Likert scales. A standardised, self-administered questionnaire was used to gather information about sources of information consulted by doctors working in the four hospitals. They could either complete the questionnaire on a Word template that was emailed to them as an attachment or as a printed hard copy, or as a link to a Google Forms created questionnaire that was shared to them using email.

### 2.4. Statistical Analysis

Data were captured and coded in Microsoft Excel and exported to STATA 17.0 and SPSS version 26 for analysis. Numerical data were explored for normality using the Shapiro–Wilk test. Numerical data were not normally distributed and were therefore summarised using the median and interquartile range (IQR). Frequency tables, graphs, and percentages were used to summarise categorical variables. Where data were missing, data were analysed using complete case analyses.

### 2.5. Ethical Considerations

This study obtained ethical clearance from the Human Research Ethics Committee of the Faculty of Health Sciences at the University of the Witwatersrand and the Faculty of Health Sciences of Walter Sisulu University Human Research Ethics and Biosafety Committee. Permission to access the identified research sites was sought from the Eastern Cape, Mpumalanga, Limpopo, and Northern Cape Provincial Health Research Committees (PHRC) and specific health facilities. The research abided by the four ethical principles of autonomy, beneficence, non-maleficence and justice. Participants’ anonymity was maintained and each participant offered informed, voluntary, written consent before participation. Respondents had the right to withdraw at any stage during the study. Participants were informed that they would not receive any incentives and that they would not be prejudiced if they chose not to participate or withdraw.

## 3. Results

Two-hundred and twenty-one (221) doctors were surveyed in the four hospitals; 54.3% (120/221) were male; 39.4% (87/221) and 15.8% (35/221) worked at NMAH and RMSH, respectively; 52.9% (108/204) were from one of five clinical departments (Emergency Department, General Surgery, Orthopaedics, Obstetrics and Gynaecology, and Internal Medicine). Age was only available for 62.0% (137/221) of the participants; the median age was 34 years (IQR = 11 years) and 85.4% (117/137) were below 45 years. The socio-demographic characteristics are summarised in Table 1.

Almost half of respondents always worked in the outpatient’s department (OPD) during the day (49.5%, 107/216); 43.9% (93/212) often saw patients in the Emergency department (ED) during the day; 50.2% (107/213) often saw patients in the ED after hours; more than two-thirds always performed ward rounds (68.2%, 148/217); and 55.5% (117/211) seldomly reviewed patient complaints (Table 2).

On sources of information, 53.8% (114/212) often or always spoke to people outside the hospitals; 69.4% (150/216) and 70.3% always consulted other doctors or colleagues, respectively; 62.0% (134/216), 52.1% (111/213), 57.5% (123/214) and 40.4% (88/218) seldomly read newspapers, reference books kept in the library, used any library books, or used hospital computers to access the internet, respectively (Table 3). Further shown in Table 3 is that attendance of training workshops was seldom for 44.4% (95/214), 62.9% (134/213) and 67.3% (142/211) irrespective of whether they were organised by non-government organisations (NGOs), provincial health office or district health office, respectively.

Trustworthiness (84.5%, 180/213), accessibility (68.8%, 148/215), format (56.1, 119/212) and familiarity (52.1%, 111/213) with the information source were very important reasons for choosing an information source (Figure 1. Reasons for choosing information sources.

## 4. Discussion

This survey study of four South African referral hospitals ascertained and established the information sources of 221 doctors. Doctors performed diverse duties within the hospitals, ranging from outpatients department to theatre. Whilst doctors mainly obtained information from consulting their colleagues or other doctors, 31.6% (67/212) consulted individuals outside of the workplace before searching online or evaluating hard copies for evidence. Notwithstanding, information sources were preferred based on their trustworthiness, accessibility, and their format. Doctors in these hospitals performed different duties, but, their shared primary responsibility was to ensure patients’ safety and well-being.

The identified information sources are consistent with those described in the literature [26,27,28,29], where clinical guidelines, colleagues, hospital policies/manuals, mobile phones, in-service training, library books, specialist physicians, and ward rounds were preferred information sources in hospitals. But differs from that reported by Ajuwon et al. [30] where most respondents accessed internet information resources from their mobile phones. Doctors often interact with colleagues who serve as an informal and formal primary source of information in the hospital setting [16,17]. This practice has not changed despite the vast increase in online information and improved accessibility [11]. According to Smith [1], humans will remain the primary source of information, regardless of how sophisticated the sources become. However, it was found that in most studies, humans became a second choice because in many cases doctors find it more convenient to seek information from their colleagues [31].

Accessibility to information sources and time factors play a role in choosing a particular source of information [6]. As a result, colleagues are often relied upon to provide needed information since they are familiar with the context and readily available at the point of care [6]. However, relying on colleagues may lead to medical errors if the information is not verified [32]. At the very least, hospitals, especially teaching hospitals must be supplied with means for verifying the accuracy of information offered by colleagues. More so, because these hospitals have additional responsibilities of managing complex cases and that of teaching health professionals in training.

The easiest way of ensuring this is by improving access to online medical resources that meet their needs at the point of care, and by having resourced libraries within hospitals. Not only should there be network availability, but there should also be fast access connectivity, preferably a secured Wi-Fi network, and subscriptions to reliable databases and journals. It would be beneficial to introduce the intranet in the hospital wards for doctors to access information at the point of care as the need arises [28]. Through active subscriptions to open educational resources, hospitals can increase doctors’ access to information. Once access to online resources is ensured, and secured with the latest antispyware technology, hospitals should also institute mechanisms for ensuring that this internet prevents or minimises access to sites that are not relevant for doctors’ duties. This is because if left open, the internet can be abused and could end up being counterproductive.

Previous studies [5,10] identified poor information and communication technology (ICT) infrastructure, poor computer access, frequent power outages, and lack of time to be among the leading factors hindering doctors’ access to information sources in South Africa. On the contrary, some doctors have been reported to underutilise some of the available resource despite them being readily available and accessible to them [24]. Online information sources are also not fully appreciated as a potential service provided by libraries [33]. As was the case in this study, open-source peer-reviewed information was not widely used despite its availability to them.

Although it is difficult for some doctors to access and use information at the point of care because they lack ICT skills and time to learn how to use them [5,10,31]. Hence, the responsibility of Continuous Professional Development (CPD) should rest with each doctor to enhance their skills [28]. It is then the hospital’s responsibility to motivate and ensure quality patient care by instilling a clear system or policies and a mandatory and enforceable CPD point target to be met by each clinician [34]. Studies show that electronic knowledge resources have a positive impact on patient outcomes as well as knowledge and behaviour among healthcare professionals [35]. Though reasons were not explored, computers were rarely used by medical doctors as sources of information in this study. This is a situation that should be improved. Doctors should also be able to use their mobile phones to access information at the point of care if there is internet access.

Even though it is not erroneous to use colleagues as information sources, but they should not be the main source. As a starting point, clinicians can be encouraged to make use of readily available clinical guidelines and protocols within their sections [36]. Among the strategies effective in improving health professionals’ behaviour are educational materials, educational meetings, reminders, audits, and feedback [37]. It has been found that reminders designed to address specific needs are more likely to change behaviour if they require or prompt professionals’ response [37]. It is also possible to integrate the appropriate resources into each doctors’ workflows by understanding their daily information needs and information-seeking behaviours [4]. Even though hospitals can improve access and availability of resources, there must be evidence-based strategies instituted to improve uptake and change doctors’ mindsets from the status quo.

Such strategies include increasing the perceived susceptibility of doctors to the consequences of not verifying shared information through evidence-based information sources [38]. This could be performed by educating them about the benefits of consulting sources with the latest evidence in their practice, and the risks they might suffer if they do not. It is also important to increase the perceived severity of those consequences [38] by highlighting the negative outcomes that can occur if doctors do not verify information.

Credibility of an information source influences the amount of trust that the information user can build, which determines whether the information is used and how it is utilized [39]. Establishing credibility as a trustworthy information provider is a crucial aspect [39]. Since libraries provide access tools such as abstracts, indexes, and bibliographies to its clientele, it is their responsibility to support hospitals using ICT facilities to enhance resource accessibility at the point of care [40]. Through pamphlets, orientation, and user education training, libraries can train doctors on how to use library resources, providing them with skills and sources to prevent impatience [40].

To our knowledge this is the first study to investigate the clinical information sources consulted by doctors at the point of care in South African teaching hospitals. Findings from this study can be used to improve the patient care information system and supply doctors with convenient and reliable sources of information. This quantitative study is limited to information sources doctors consult to answer clinical questions in the four referral hospitals which may limit generalisability of the findings. This study is further limited by the use of a survey with mostly closed-ended questions. This limited the ability to probe and ascertain the true nature of the resources that could be available within the hospitals. However, that could be a subject of future studies. The study is also limited by the inability to compare the factors associated with the information sources based on the hospital or workstation. However, this study achieved its objective of understanding the information sources doctors consult in the selected South African referral hospitals.

## 5. Conclusions

This study on the exploration of information sources in four South African rural referral hospitals achieved its aim. Doctors mainly relied on colleagues and sometimes consulted individuals outside of the workplace before searching online or evaluating hard copies for evidence. Information sources were preferred based on their trustworthiness, accessibility, and their format and not so much on their cost. Doctors must be supported with means for corroborating the information sourced from other health professionals such as by increasing access to online resources and resourced libraries within their hospitals. Furthermore, evidence-based strategies to encourage uptake of these resources should be undertaken.

## Figures and Tables

**Figure 1 healthcare-12-00008-f001:**
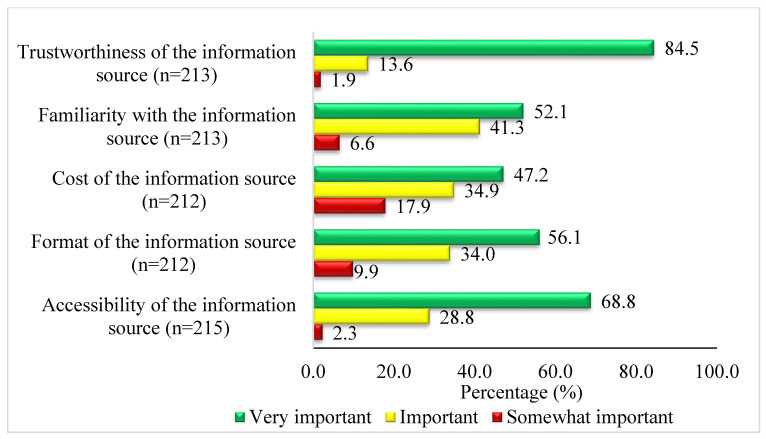
Reasons for choosing information sources.

**Table 1 healthcare-12-00008-t001:** Doctor’s demographic characteristics.

Characteristics	N = 221	(100%)
Age ^#^, years; median (IQR)	34	(11)
Age groups ^#^; years; n (%)		
25–35	72	(52.6)
36–45	45	(32.8)
46–55	14	(10.2)
56–64	6	(4.4)
Gender *; n (%)		
Male	120	(54.3)
Female	101	(45.7)
Hospital *; n (%)		
Nelson Mandela Academic Hospital	87	(39.4)
Pietersburg Hospital	54	(24.4)
Robert Mangaliso Sobukwe Hospital	35	(15.8)
Witbank Hospital	45	(20.4)
Section ^ʊ^; n (%)		
Emergency Department	35	(17.2)
General Surgery and Orthopaedics	30	(14.7)
Obstetrics and/or Gynaecology	24	(11.8)
Internal Medicine	19	(9.3)
Paediatrics	17	(8.3)
Oncology	12	(5.9)
Dermatology	12	(5.9)
Neurosurgery	8	(3.9)
Dental unit	7	(3.4)
Administration	5	(2.5)
Radiology	5	(2.5)
Other specialised surgery ^α^	5	(2.5)
Theatre, and/or Anaesthesia	5	(2.5)
ICU	4	(2.0)
Other	16	(7.8)

IQR = 75th percentile—25th percentile; ^#^ n = 137; * N = 221; ^ʊ^ n = 204; Other = (OPD, Ophthalmology, Psychiatry, Urology (n = 3); Neurology (n = 2); Nephrology and Nuclear Medicine (n = 1)); ^α^ = Plastic surgery = 2, Maxillo-facial and oral surgery = 2.

**Table 2 healthcare-12-00008-t002:** Daily duties performed.

Duties	Seldom	Often	Always	Total
See patients in outpatients during day; n (%)	31 (14.4)	78 (36.1)	107 (49.5)	216 (100.0)
See patients in emergency department during the day; n (%)	64 (30.2)	93 (43.9)	55 (25.9)	212 (100.0)
See patients in emergency department after hours; n (%)	51 (23.9)	107 (50.2)	55 (25.8)	213 (100.0)
Perform ward round to see patients admitted in the ward; n (%)	38 (17.5)	31 (14.3)	148 (68.2)	217 (100.0)
I work in the theatre; n (%)	95 (47.7)	54 (27.1)	50 (25.1)	199 (100.0)
Perform minor procedures; n (%)	40 (18.6)	88 (40.9)	87 (40.5)	215 (100.0)
Request and interpret blood investigations; n (%)	10 (4.6)	24 (11.0)	185 (84.5)	219 (100.0)
Request and interpret x-ray investigations; n (%)	9 (4.1)	32 (14.7)	176 (81.1)	217 (100.0)
Prescribe treatment for sick patients; n (%)	9 (4.1)	20 (9.2)	188 (86.6)	217 (100.0)
Educate patients about their illness; n (%)	8 (3.7)	31 (14.3)	178 (82.0)	217 (100.0)
Give treatment to patients; n (%)	35 (16.4)	25 (11.7)	154 (72.0)	214 (100.0)
Review progress of patients on treatment; n (%)	12 (5.6)	27 (12.5)	177 (81.9)	216 (100.0)
Teach health workers and health sciences students; n (%)	56 (26.2)	74 (34.6)	84 (39.3)	214 (100.0)
Review mortality statistics; n (%)	72 (33.6)	86 (40.2)	56 (26.2)	214 (100.0)
Conduct folder reviews; n (%)	89 (42.6)	70 (33.5)	50 (23.9)	209 (100.0)
Review complaints made by patients; n (%)	117 (55.5)	48 (22.7)	46 (21.8)	211 (100.0)

**Table 3 healthcare-12-00008-t003:** Preferred information sources.

Where, Does the Doctor Look to Get Information to Meet His/Her Needs?	Seldom	Often	Always	Total
	n (%)	n (%)	n (%)	n (%)
Talk to colleagues	2 (0.9)	63 (28.8)	154 (70.3)	219 (100.0)
Consult other doctors	12 (5.6)	54 (25.0)	150 (69.4)	216 (100.0)
Talk to people outside of work	98 (46.2)	47 (22.2)	67 (31.6)	212 (100.0)
Read newspaper	134 (62.0)	54 (25.0)	28 (13.0)	216 (100.0)
Use computer at work to access internet	88 (40.4)	50 (22.9)	80 (36.7)	218 (100.0)
Use reference books kept in the hospital	111 (52.1)	51 (23.9)	51 (23.9)	213 (100.0)
Use protocols/guidelines kept in the ward or pocketbook	46 (21.0)	61 (27.9)	112 (51.1)	219 (100.0)
Consult hospital policy manual	91 (41.9)	61 (28.1)	65 (30.0)	217 (100.0)
Use any library books	123 (57.5)	53 (24.8)	38 (17.8)	214 (100.0)
Attend seminars run in the hospital	57 (26.5)	81 (37.7)	77 (35.8)	215 (100.0)
Attend training workshops organised by NGO	95 (44.4)	72 (33.6)	47 (22.0)	214 (100.0)
Attend training workshops organised by provincial office	134 (62.9)	48 (22.5)	31 (14.6)	213 (100.0)
Attend training workshops organised by district office	142 (67.3)	43 (20.4)	26 (12.3)	211 (100.0)

## Data Availability

All data used in this study will be available from the first author upon reasonable request.

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
