# Peer review of "Exploring the Information Sources Consulted by Doctors at the Point of Care in Four Selected South African Referral Hospitals"

_healthcare, 2023, doi:10.3390/healthcare12010008_

Round 1

Reviewer 1 Report

Comments and Suggestions for Authors

Thanks for your valuable reading. A few comments can be effective in improving your work

Questionnaire information needs more explanation

 If you refer to a validated questionnaire in another article, if it is standardized in this study, explain it.

Several questions are considered for each part of the questionnaire

  Likert scale, how many options are chosen to answer?

State the ethical considerations of the study.

  Describe the research process and the process of distributing and completing the questionnaire.

 The classification of people in different age groups is not equal to 221. check it

Table 1 should be checked again.

Reference 40 is not in the references

Author Response

Reviewer 1 Response

Thank you for your comments.

  1. Comment: The questionnaire information needs more explanation.

Response: Thank you for the comment. Additional information has been added to the manuscript, Lines 143 -159:

The questionnaire (Appendix A) was adapted from a validated instrument that was previously used to ascertain the information behaviour of nurses and doctors of one rural South African district. A validated and standardised structured self-administered questionnaire was used, and asked questions on demographic characteristics, responsibilities, preferred information sources and reasons for choosing an information source. The Cronbach’s alpha had good internal consistency (0.865) for the 25 items assessed on the Likert scale. A standardised, self-administered questionnaire was used to gather information about sources of information consulted by doctors working in the four hospitals. They could either complete the questionnaire on a Word template that was emailed to them as an attachment or as a  a link to a Google Forms-created questionnaire that was shared to them using email or WhatsApp.

  1. Comment: If you refer to a validated questionnaire in another article, if it is standardised in this study, explain it.

Response: Thank you for the comment. The point has been clarified to explain that:

The questionnaire (Appendix A) was adapted from a validated instrument that was previously used to ascertain the information behaviour of nurses and doctors of one rural South African district (Chitha, N., 2017. Information behavior of medical doctors and professional nurses in selected hospitals of OR Tambo Health District, Eastern Cape Province, South Africa (Doctoral Thesis)) (Line 143-146).

  1. Comment: Several questions are considered for each part of the questionnaire

Response: Thank you for the comment. Line 147-149 expresses that the questionnaire, “…asked questions on demographic characteristics, responsibilities, preferred information sources and reasons for choosing an information source”.

  1. Comment: Likert scale, how many options are chosen to answer?

Response: Thank you for the comment. Line 151 now clarifies that two Likert scales were used. One being a three-item and the other a five item scale.

  1. Comment: State the ethical considerations of the study.

Response: Thank you for the comment. Ethical considerations have been added in lines 171-187.

This study obtained ethical clearance from the Human Research Ethics Committee of the Faculty of Health Sciences at the University of the Witwatersrand and the Faculty of Health Sciences of Walter Sisulu University Human Research Ethics and Biosafety Committee. Permission to access the identified research sites was sought from the Eastern Cape, Mpumalanga, Limpopo, and Northern Cape Provincial Health Research Committees (PHRC) and specific health facilities. The research abided by the four ethical principles of autonomy, beneficence, non-maleficence and justice. Participants’ anonymity was maintained and each participant offered informed, voluntary, written consent before participation. Respondents had the right to withdraw at any stage during the study. Participants were informed that they would not receive any incentives and that they would not be prejudiced if they chose not to participate or withdraw.

  1. Comment: Describe the research process and the process of distributing and completing the questionnaire.

Response: Thank you for the comment. Information on how the questionnaire was distributed has been added, Lines 153 – 161.

They could either complete the questionnaire on a Word template that was emailed to them as an attachment or as a printed hard copy, or as  a link to a Google Forms-created questionnaire that was shared to them using email or WhatsApp.

  1. Comment: The classification of people in different age groups is not equal to 221. check it Table 1 should be checked again.

Response: Thank you for the comment. The first two lines of Table 1 (Age#, years; median (IQR) and (Age groups#; years; n (%)) have a key (#) whose footnote clarifies that #n=137. So, there is therefore no error in Table 1.

  1. Comment: Reference 40 is not in the references

Response: The response has been updated on the manuscript Lines 467 - 469

Okeke OC, Ezu SG, Eze JU, Asogwa GE. Status of medical library resources and services in teaching hospitals in Enugu State, Nigeria: Implications for quality health care services. International journal of Knowledge content development & technology. 2017 Jun 1;7(2).

Reviewer 2 Report

Comments and Suggestions for Authors

File attached.

Comments on the Quality of English Language

Minor editing of English language required

Author Response

Reviewer 2 Response

Title: Understanding the information sources consulted by doctors at the point of care in four selected South African referral hospitals.

  1. Comment: Overall: It is a commendable piece of research on sources of information.

Response: Thank you for your comment.

  1. Comment: Title: It is suggested the term “Understanding” with “Exploring” or something else that has scientific connotation.

Response: Thank you for the comment. The title has been revised to: Exploring the information sources consulted by doctors at the point of care in four selected South African referral hospitals

  1. Comment: Abstract: It is good. Keywords are mentioned.

Response: Thank you for the comment.

  1. Comment: Introduction: The introduction aligns with the objectives. The introduction is well written.

Response: Thank you for the comment.

  1. Comment: The Authors have well mentioned the critical updated sources of information at the point of patient care.

Response: Thank you for the comment.

  1. Comment: Method & Design: The research approach used for this study to realise the objectives of the study is appropriate.

Response: Thank you for the comment.

  1. Comment: Why only 4 provinces were selected out of 9 SA? It is good that the researcher selected one referral hospital from each province.

Response: Thanks for the comment. This has been clarified to explain that, “Due to resource limitations, four South African rural provinces (Limpopo (LP), Eastern Cape (EC), Northern Cape (NC) and Mpumalanga (MP) were randomly selected for this study” (line 132-134).

  1. Comment: Could you provide statistics on the number of hospitals in each province?

Response: The response has been updated on the manuscript Lines 111 - 125

Private and public health systems coexist side by side in South Africa. The great majority of people are served by the public health sector. This study focused on Tertiary and Central hospitals. The National Department of Health, province health departments, and local health departments each have different levels of authority and responsibility for providing services. Over 200 private hospitals and over 400 governmental hospitals are available. The largest regional hospitals are directly managed by the provincial health departments. District management oversees primary care facilities and District hospitals. The 10 Central hospitals are directly managed by the Department of Health at the national (federal) level. Among the nine provinces, the Eastern Cape has 91 public hospitals, of which one is a Central hospital and three are Tertiary hospitals; Limpopo has 42 public hospitals of which one is a Tertiary hospital; Mpumalanga has 33 public hospitals of which two are Tertiary hospitals;  and the Northern Cape has 16 public hospitals of which one is a Tertiary hospital. Given these, one Tertiary hospital was purposively chosen from each of the three provinces, with the only Central hospital selected in the Eastern Cape province.

  1. Comment: In Line numbers: 118 and 119 it is stated, “The sample size has been previously discussed by Chitha 25 in the literature, a brief review of those discussions is provided here.” It is suggested that the researchers should extract the information from the source and write in words and present the reference. Furthermore, the authors did not provide a brief review of those discussions.

Response: Thank you for the comment. The reference being referred to is a published protocol of this study. The researchers therefore felt it important to not self-plagiarise but to also not repeat detail that can easily be sourced from the published protocol. But instead limit the body of work of this research to important detail so that readers’ attention wouldn’t be lost.

  1. Comment: In line number: 123, it is stated, “A validated and standardised structured self-administered questionnaire was used..”Was this questionnaire developed by someone else or the authors of this paper? If it is by the authors of this paper, how did you develop the questionnaire (process of validation and standardization)?

Response: The response has been updated on the manuscript Lines 149 -156

The questionnaire (Appendix A) was adapted from a validated instrument that was previously used to ascertain the information behaviour of nurses and doctors of one rural South African district. A validated and standardised, structured self-administered questionnaire was used, and asked questions on demographic characteristics, responsibilities, preferred information sources and reasons for choosing an information source.

  1. Comment: Line number: 125 & 126, Further it is stated, “The Cronbach’s alpha had good internal consistency (0.865) for the 25 items assessed on the Likert scale.”

Response: Thank you for the comment.

  1. Comment: There are more than 25 items in the questionnaire, why not Cronbach’s alpha for the whole questionnaire?

Response: Cronbach’s alpha can only be used for a Likert scale. So, this was limited to the 25 relevant questions. It would for instance not be possible to get a Cronbach’s alpha for a respondent’s age or sex, etc.

  1. Comment: It is good that normality of the data was tested. Thus, consider use Chi Square to find the significant differences in the distribution rather than simple presentation of percentage.

Response:

  1. Comment: It is good that ethical approval of the research is stated.

Response: Thank you for the comment.

  1. Comment: Results: All results have been reported correctly. Suggested data analysis may bring changes in the presentation of results.

Response: Thank you for the comment. This manuscript is meant to be descriptive. A different manuscript that is under review is more analytical with p-values and other relevant statistics.

  1. Comment: Discussion: The discussion section is well presented with appropriate references.

Response: Thank you for the comment.

  1. Comment: Conclusion: The conclusion has clarity.

Response: Thank you for the comment.

  1. Comment: Limitations of study: The researchers have outlined the limitations of the current research.

Response: Thank you for the comment.

  1. Comment: All the following sections have been satisfactorily answered: Acknowledgments, Author Contributions, Conflicts of Interest, Funding, Data Availability Statement

Response: Thank you for the comment.

Reviewer 3 Report

Comments and Suggestions for Authors

Dear Authors,

I have read your manuscript, which raises the important issue of the sources of doctors' knowledge. As we know, medical staff are obliged to continue their education per all professional life. It is also important where the knowledge comes from because it should be based on scientific evidence.

The group of respondents is large and there is a positive element of your research.

However, I have a few comments about the manuscript:

1.       I believe that the topic should be changed to: sources of doctors' knowledge... because that is what you are asking about in the study

2.       I did not find a clearly articulated purpose of the research

3.       Line 109: The methods used in this study have been described by Chitha – I do not know this author and do not have access to this study. Please give me specific details

4.       Participants and Setting – how did you conduct this research? Did you hand out the questionnaires in person? How were they returned? How did you ensure anonymity?Statistical analysis – what level of significance did you adopt?

5.       Table 2. Daily duties of a performer – why do we need this information? How are they related to the topic of the work? I believe that this data does not add anything to the study

6.       Table 3. Preferred information sources - Read newspaper – which newspapers: professional or popular science or other?

7.       Figure 1. Reasons for choosing information sources – can be removed, it is more a decoration of the manuscript than in-depth information. Instead, it would be better to show which respondents use selected sources and what guides them when choosing them. This information should be further developed.

8.       I don't see the results of statistical analyses. The Results include only descriptive statistics.

9.       Lines: 141, 151, 158 - Error! Reference sources not found. I guess that's a mistake?

      References – 1/3 of the literature is more than 10 years old!

The study is very general, it would be worth further analysis and drawing appropriate conclusions.

Author Response

Title: Understanding the information sources consulted by doctors at the point of care in four selected South African referral hospitals

Reviewer 3 Response

General Comment: I have read your manuscript, which raises the important issue of the sources of doctors'

knowledge. As we know, medical staff are obliged to continue their education per all

professional life. It is also important where the knowledge comes from because it should be

based on scientific evidence. The group of respondents is large and there is a positive element of your research. However, I have a few comments about the manuscript:

Response: Thank you for the kind words, Noted.

  1. Comment: I believe that the topic should be changed to: sources of doctors' knowledge... because that is what you are asking about in the study.

Response: The title has been revised to: “Exploring the information sources consulted by doctors at the point of care in four selected South African referral hospitals”

  1. Comment: I did not find a clearly articulated purpose of the research

Response: The response has been updated on the manuscript. In addition to having stated that: “This study therefore sought to determine the clinical information sources consulted by referral hospital-based doctors, for making decisions at the point of patient care in four South African provinces” (Line 103-106). Line 106-107 has been added to state that: “This information will be shared with policymakers to enhance the strengthening of health systems and teaching hospitals in South Africa”.

  1. Comment: Line 109: The methods used in this study have been described by Chitha – I do not know this author and do not have access to this study. Please give me specific details

Response: Thank you for the comment. There is a reference to the study’s protocol by this author, that we can’t repeat in this manuscript given the limited wordcount and the high possibility of plagiarism. We have otherwise added a bit more detail for clarity in lines 110-118:

This study used a cross-sectional survey design using a quantitative methodology. Hospitals and provinces were determined by simple random selection. A validated questionnaire will be given to doctors who were chosen through stratified random selection. A standardised, self-administered questionnaire will be used in the investigation. Robert Mangaliso Sobukwe Hospital (Northern Cape province), Pietersburg Hospital (Limpopo province), Witbank Hospital (Mpumalanga province), and Nelson Mandela Academic Hospital (Eastern Cape province) were the sites of choice.

  1. Comment: Participants and Setting – how did you conduct this research? Did you hand out the questionnaires in person? How were they returned? How did you ensure anonymity?

Statistical analysis – what level of significance did you adopt?

Response: The response has been updated on the manuscript Lines 150 – 166 to clarify that:

The questionnaire (Appendix A) was adapted from a validated instrument that was previously used to ascertain the information behaviour of nurses and doctors of one rural South African district. A validated and standardised, structured self-administered questionnaire was used, and asked questions on demographic characteristics, responsibilities, preferred information sources and reasons for choosing an information source. The Cronbach’s alpha had good internal consistency (0.865) for the 25 items assessed on the three and five item Likert scales. A standardised, self-administered questionnaire was used to gather information about sources of information consulted by doctors working in the four hospitals. They could either complete the questionnaire on a Word template that was emailed to them as an attachment or as a printed hard copy, or as  a link to a Google Forms-created questionnaire that was shared to them using email or WhatsApp.

This was a descriptive study, so there was no hypothesis testing or estimation that necessitated the use of p-values or confidence intervals.

  1. Comment: Table 2. Daily duties of a performer – why do we need this information? How are they related to the topic of the work? I believe that this data does not add anything to the study

Response: This is actually very important as it firstly demonstrates the heterogeneity of participants and further show that the information sources were almost similar regardless of their daily duties.

  1. Comment: Table 3. Preferred information sources - Read newspaper – which newspapers:

            professional or popular science or other?

Response: Thank you for the comment. The questionnaire does not specify the types of newspapers, the frequency or the language. Media is known to at times disseminate reliable health information. This question wanted to establish the frequency of reliance on printed media for their professional information.

  1. Comment: Figure 1. Reasons for choosing information sources – can be removed, it is more a decoration of the manuscript than in-depth information. Instead, it would be better to show which respondents use selected sources and what guides them when choosing them. This information should be further developed.

Response: Thank you for the comment. This Figure (Figure 1) actually points out the three most important reasons for choosing information and these are compared to literature findings in the Discussion section. We therefore disagree that this is only used for cosmetics.

  1. Comment: I don't see the results of statistical analyses. The Results include only descriptive Statistics.

Response: The Results use descriptive statistics as described in the Statistical analysis section. We suspect that the Reviewer meant to say that they don’t see results of hypothesis testing instead of saying there are no “…results of statistical analyses…”. Percentages and frequencies are a result of the statistical analyses. We have more body of work on this research that is being reviewed and has a focus on hypothesis testing and comparing different groups with p-values (hypothesis testing) and confidence intervals (estimations). The purpose of this study was to merely understand/explore the sources of information and we believe that we have done justice to the research question.   

  1. Comment: Lines: 141, 151, 158 - Error! Reference sources not found. I guess that's a mistake?

Response: Thank you for the comment. Yes, it’s a mistake that has since been rectified.

  1. Comment: References – 1/3 of the literature is more than 10 years old!

Response: The response has been updated on the manuscript Lines 362 -365 and the references have been updated.

  1. Gentry S, Badrinath P. Defining health in the era of value-based care: lessons from England of relevance to other health systems. Cureus. 2017 Mar 6;9(3).
  2. Neighbour R. The inner consultation: how to develop an effective and intuitive consulting style. CRC press; 2018 Feb 6.
  3. Rodziewicz TL, Hipskind JE. Medical error prevention. StatPearls. Treasure Island (FL): StatPearls Publishing. 2020 Jan.

  1. The study is very general, it would be worth further analysis and drawing appropriate conclusions

Response: Thank you for the comment. The first line of the conclusions has been updated to align with the revised title. This changes the meaning of the conclusions and ensures that the conclusions are fully aligned with the study objectives.

Round 2

Reviewer 1 Report

Comments and Suggestions for Authors

Thanks to the editor

The manuscript can be accepted with the corrections made.

Author Response

Thank you for the comment

Reviewer 3 Report

Comments and Suggestions for Authors

Dear Authors,

Thank you for considering my suggestions and clarifying the issue that were unclear to me. The manuscript now reads with greater understanding.

I miss references to figure and tables in the text. I still feel dissatisfied with Table 2. It is undestood that doctors have their duties, but ho wis this reflected in your research results? You do not describe or utilize these date beyond general statements in line 188-192 and 216-217.

What does the color red signify in Figure 1?

Author Response

Reviewer 3

Dear Authors,

Comment: Thank you for considering my suggestions and clarifying the issue that were unclear to me. The manuscript now reads with greater understanding.

Response: Noted. Thank you.

Comment: I miss references to figure and tables in the text.

Response: Thank you for the comment. The references have now been added.

Comment: I still feel dissatisfied with Table 2. It is undestood that doctors have their duties, but ho wis this reflected in your research results? You do not describe or utilize these date beyond general statements in line 188-192 and 216-217.

Response: Thank you for the comment. We have now added a sentence in the Discussion:

Doctors performed diverse duties within the hospitals, ranging from outpatients department to theatre. (Line 219-220).

We have further added a limitation:

The study is also limited by the inability to compare the factors associated with the information sources based on the hospital or workstation. (Line 311-313).

Comment: What does the color red signify in Figure 1?

Response: Thank you for the comment. The Figure size was compressed during the Editing process and hid some of the detail but that has since been corrected to reflect that red stands for “Somewhat important”